# Quantifying Conformational Isomerism in Chain Molecules by Linear Raman Spectroscopy: The Case of Methyl Esters

**DOI:** 10.3390/molecules26154523

**Published:** 2021-07-27

**Authors:** Maxim Gawrilow, Martin A. Suhm

**Affiliations:** Institut für Physikalische Chemie, Georg-August-Universität Göttingen, Tammannstr. 6, 37077 Göttingen, Germany; maxim.gawrilow@chemie.uni-goettingen.de

**Keywords:** Raman intensity, conformational isomerism, chain folding, jet cooling, rotational band contour, esters

## Abstract

The conformational preferences of the ester group have the potential to facilitate the large amplitude folding of long alkyl chains in the gas phase. They are monitored by Raman spectroscopy in supersonic jet expansions for the model system methyl butanoate, after establishing a quantitative relationship with quantum–chemical predictions for methyl methanoate. This requires a careful analysis of experimental details, and a simulation of the rovibrational contours for near-symmetric top molecules. The technique is shown to be complementary to microwave spectroscopy in quantifying coexisting conformations. It confirms that a C−O−C(=O)–C–C chain segment can be collapsed into a single all-trans conformation by collisional cooling, whereas alkyl chain isomerism beyond this five-membered chain largely survives the jet expansion. This sets the stage for the investigation of linear alkyl alkanoates in terms of dispersion-induced stretched-chain to hairpin transitions by Raman spectroscopy.

## 1. Introduction

Vibrational spectroscopy is best for accurately measuring vibrational fundamental wavenumbers, and it often allows for the unambiguous identification of molecules [1]. If the spectra cover a wide spectral range and include some control of the conformational composition, the identification of individual conformations is also possible [2]. Being able to quantify this conformational composition of a compound would be of added value, because it allows for conclusions to be drawn about the energy ordering of molecular conformations [3,4]. This requires the quantitative experimental determination of, at least, relative intensities in the gas phase and a reliable relationship between theoretical and experimental intensities. These two requirements are currently met more robustly and frequently by infrared spectroscopy than by Raman spectroscopy [5,6,7]. However, Raman spectroscopy is typically more powerful in terms of spectral coverage and conformational resolution, due to the accessibility of low-frequency vibrations and the frequent occurrence of sharp *Q*-branch transitions. Therefore, the present contribution discusses the different aspects, challenges and limitations of using linear Raman spectroscopy for the quantification of conformational isomers.

Short-chain esters were chosen because their longer chain homologues are thought to be ideal candidates for the characterisation of a chain-folding isomerism. In contrast with simple alkanes, where such dispersion-driven chain-folding has already been studied in detail [4], they offer a low barrier folding hinge, due to the conformational properties of the ester group. Similar to phenyl alkanes [8], this facilitates the folding process and makes it accessible for shorter chain lengths.

Here, methyl methanoate (methyl formate), as the simplest alkyl ester, is used to explore the general performance and difficulties in the comparison of theoretical and experimental Raman intensities. These insights are then used to quantify the conformational isomerism in methyl butanoate and compare the results to recent findings, using combined broad-band and high-resolution microwave spectroscopy [9], where, instead of theoretical Raman transition polarisabilities, theoretical dipole moment components were required. This paves the way for the semi-quantitative identification of folded isomers in longer alkyl chains with mid-chain ester group placement as a function of chain length, to be published elsewhere.

Some of the main challenges are illustrated in Figure 1, which shows the gas-phase room-temperature Raman spectrum of the monoconformational (vide infra) methyl methanoate, recorded in overlapping sequences using a state-of-the-art Raman spectrometer developed for extremely dilute supersonic jet spectra [10]. Although the signal-to-noise ratio is about two orders of magnitude better than in a previous spectrum reported in the literature [11], it is difficult to determine relative integrated signal intensities for several vibrations, because they overlap with each other, at least in their rotational wings. There are very few regions where the spectral baseline drops to zero between the signals (see right inset in Figure 1 for the connecting wings). Note that methyl methanoate is far smaller than the target esters; therefore, spectral complexity due to conformational diversity is greatly reduced, although the width of the rovibrational wings is particularly large. Any attempt to accurately quantify ester conformations is bound to fail in such a situation, unless the rovibrational contour can be simulated reliably. In one specific example, some signals have an integration error due to baseline uncertainty on the order of only 10%. Many others, even if grouped together, carry a much larger uncertainty and, therefore, cannot be adequately quantified (ESI, Appendix A).

Furthermore, comparison with theoretical intensities is typically made at 0 K, where only the rovibrational ground state is populated. At room temperature, even for this small model ester, the rotational ground state population is less than 0.01%, and the vibrational ground state population is only about 30%. Even if the theoretical predictions are correct at 0 K, they may still not reflect the experimental intensities. It appears indispensable to move to a supersonic jet experiment, where the compound of interest is mixed with a large excess of a carrier gas and expanded more or less isentropically into vacuum, collisionally converting internal energy into a directed translation [12]. In this process, rotational cooling is very pronounced and vibrational cooling is at least partial. This is illustrated in the left inset of Figure 1, where a jet-cooled spectrum of the intense ester linkage vibration (vide infra) is plotted, together with the gas phase spectrum. The intensity scaling is chosen such that the jet spectrum does not cross the gas-phase spectrum. In this way, the fractional intensity of this strong signal (almost two orders of magnitude lower) reflects the degree of cooling of vibrational and, in particular, the rotational states achieved in the jet spectrum, at least in a qualitative way. Most of the additional *Q*-branch intensity at lower wavenumbers is due to transitions from thermally excited states. Much of this problematic hot band structure is removed in the jet and a comparison to theoretical intensities appears more promising. This allows for conformational relaxation studies in the supersonic jet, which provide further insights into relative energy ordering.

For long-chain esters with conformational isomerism, this jet-cooling effect is not enough to render an intensity comparison based on integrated intensities feasible, because one can still observe significant overlap between isomeric signals, which now become important. This is illustrated in Figure 2 for a small section of the jet-cooled spectrum of methyl butanoate, where two different conformations contribute. Therefore, in a second step, the present contribution attempts to develop an assessment based on peak intensity rather than integrated intensity. The hope is that prominent *Q*-branches allow for the detection and approximate quantification of contributing conformers in favourable spectral regions. For this to work, a semi-quantitative simulation of rotational band contours of cold Raman spectra appears to be indispensable and shall be attempted in the limit of near-symmetric tops. Note that it is sufficient to push the contour simulation error below the error expected from the quantum chemical harmonic approximation, because the latter is currently difficult to overcome in terms of theoretical Raman scattering cross sections [5]. In this respect, Raman spectroscopy still lags behind infrared spectroscopy, although both pose some non-trivial challenges for low-frequency modes [13,14]. The situation is often more favourable for harmonically predicted wavenumbers. Figure 2 shows that the anharmonic shift in the fundamentals can be sufficiently systematic to allow for an assignment of the conformations. Only quantification based on intensity remains challenging. In the chosen example, it is difficult to judge whether conformer 1 or conformer 2 is more dominant in the jet expansion, due to the spectral overlap and the theoretical intensity error. A similar observation has been reported for the Raman spectrum of liquid and solid methyl butanoate [6]. The computational models that were feasible almost 20 years ago profited from systematic anharmonic shifts and allowed for the qualitative assignment of conformers, but the calculated Raman intensities were too erratic for any quantitative analysis.

The declared goal of this work is to carefully analyse which uncertainty factors have to be taken into account from the experimental and rotational simulation side to enable a conformational quantification based on theoretical harmonic Raman scattering cross sections. Based on recent developments towards anharmonic Raman intensities [5], an average error of 20% of the harmonic approximation in relative Raman scattering intensities of Raman-active fundamentals can be roughly expected. Evidently, the effects can be much larger, particularky for overtones and combination transitions, which are completely forbidden in a double harmonic approximation [15]. Our aim would be to push the experimental uncertainties below, or at least close to, this 20% threshold for fundamentals. This is explored for the monoconformational methyl methanoate in comparison to theory and previous experiments [11], before it is tested for conformational relaxation in methyl butanoate, which has already been characterised by microwave spectroscopy [9].

Our data show that, despite the many aspects which have to be considered when combining theoretical and experimental Raman intensities, it is possible to quantify different conformations of flexible chain molecules in supersonic jets based on such intensity information. This allows for the monitoring of alkyl ester folding processes as a function of chain length, to learn more about the London dispersion forces which drive them.

## 2. Results

### 2.1. Systematic Error Sources in Experimental Intensities

We start with a list of error sources which have to be considered for quantitative Raman jet spectroscopy, mostly exemplified for methyl methanoate.

#### 2.1.1. Aggregation Effects

The price of the low population of excited rovibrational states in jet expansions is the partial aggregation of molecules into weakly bound dimers and oligomers, depending on how close to the supersonic nozzle exit the molecules are probed, and the strength of their dilution in the carrier gas. Similar to macroscopic phase changes, this can affect the relative intensities of vibrational bands, although typically less for Raman spectroscopy than for infrared spectroscopy, particularly when hydrogen bonds are involved [16]. Figure 3 illustrates the effect for methyl methanoate, most prominently for the ν(C=O) stretching mode near 1755 cm^−1^. Instead of a single fundamental band of the isolated molecule, several downshifted signals appear for higher concentrations in the uniform carrier gas, and actually become dominant. Dilution of the most concentrated gas mixture (about 3% in helium) by a factor of ~50 removes most of these aggregation features. The last dimer traces (second spectrum from the bottom, two weak signals at 1727 and 1732 cm^−1^) are lost by probing the expansion closer to the nozzle (bottom trace), albeit at the prize of warming the molecules and broadening their band contours, because the collisional cooling is less complete. The last persistent side signal at 1716 cm^−1^ shows the same scaling with concentration as the main ν(C=O) signal, and is attributed to some combination of the monomer, likely stealing intensity from the fundamental. For a reliable monomer intensity quantification, a compromise has to be made between having low dimer features, narrow rotational structure and good signal-to-noise ratio. Judging by the ν(C=O) stretching mode, the third spectrum from the bottom (bold trace) was chosen for further analysis. Rewardingly, the total integral in the ν(C=O) range was quite proportional to the concentration in this range, indicating that the integrated spectral visibility is not affected substantially by the aggregation state for this mode.

The signals below 1500 cm^−1^ are less sensitive to aggregation, as are low-frequency (<700 cm^−1^) and ν(C–H) modes (see ESI, Appendix A). Therefore, an acceptable choice for ν(C=O) is also expected to be appropriate for most other spectral ranges, considering that the Raman intensity per molecule will only weakly depend on aggregation in many cases. Furthermore, ν(C=O) is identified as a suitable aggregation-tracking mode for longer chain esters. The addition of up to 20% argon to the gas mixture to further narrow the rotational band contours was counterproductive, because it did not increase rotational cooling, but instead enhanced aggregation, and thus required higher dilution to be monomer-dominated.

#### 2.1.2. Day-to-Day Reproducibility, Statistical Noise and Impurities

An important reason not to choose the highest dilution is the signal-to-noise ratio and reproducibility of the spectra. In order to evaluate the latter, the third-lowest spectrum in Figure 3 was recorded twice on different days, and the lowest five traces were compared in terms of relative intensity (in the absence of an internal standard), to see whether dimerisation or reproducibility is more critical. As shown in the ESI (Appendix A), the statistical error (from noise level analysis as described in Section 3), the possible distortion due to dimerisation, and the day-to-day reproducibility are comparable in size for these diluted spectra. Even the weak signal at around 1580 cm^−1^, which overlaps with trace air and water impurities ($H_{2}O$ bending fundamental at 1595 cm^−1^ [17]), is not affected substantially. Overall, the error due to dimerisation and reproducibility issues is estimated below 1% of the most intense signal in a given spectral window covered by the CCD camera. In the absence of more specific quantifications of intensity distortions, such as difference spectra under varying conditions, we transfer this somewhat ad hoc absolute error bar to all signals. It should also cover water and methanol trace impurities which are detectable in the OH stretching range, with little spectral effect <1800 cm^−1^. It is of the same order as the more easily quantifiable integration error, and we add it to the latter as a conservative estimate of such non-statistical effects.

#### 2.1.3. Error Due to Angle of Observation

The scattered light is collected not only at exactly 90°, but over a certain solid angle of collimation with the employed camera objective. This affects different polarisation directions to a different extent. The effect is typically neglected, as in previous work in our group. Classically, the scattering activity for orthogonal incident laser polarisation $\left({⊥}^{i}\right)$ and orthogonal polarisation of scattered light $\left({⊥}^{s}\right)$ w.r.t. the scattering plane is $A({⊥}^{i},{⊥}^{s})$$=45{a_{k}^{\prime }}^{2}+4{\gamma_{k}^{\prime }}^{2}$, with $a^{\prime }$ and $\gamma^{\prime }$ being the isotropic and anisotropic transition polarisability invariants. For at least one parallel component $A\left({⊥}^{i},{\|}^{s}\right)=A\left({\|}^{i},{⊥}^{s}\right)={A(\|}^{i},{\|}^{s})=3{\gamma_{k}^{\prime }}^{2}$ [18]. At our setup, without a polarising filter in the scattered beam, the recorded intensity is proportional to $A\left({⊥}^{i},{⊥}^{s}\right)+A\left({⊥}^{i},{\|}^{s}\right)$.

For our collection geometry, these numbers are changed to $A\left({⊥}^{i},{⊥}^{s}\right)=45{a_{k}^{\prime }}^{2}+4.1133{\gamma_{k}^{\prime }}^{2}$; $A\left({⊥}^{i},{\|}^{s}\right)=0.0423{a_{k}^{\prime }}^{2}+3.1142{\gamma_{k}^{\prime }}^{2}$; $A\left({\|}^{i},{⊥}^{s}\right)=1.6996{a_{k}^{\prime }}^{2}+3.1511{\gamma_{k}^{\prime }}^{2}$; ${A(\|}^{i},{\|}^{s})=1.6171{a_{k}^{\prime }}^{2}+3.1492{\gamma_{k}^{\prime }}^{2}$. A detailed derivation is provided in the ESI (Section S3.2). The angular correction for the scattering activity thus strongly depends on the vibration-specific ratio between the transition polarisability components. If one compares a reasonably polarised mode ($a_{k}^{\prime }/\gamma_{k}^{\prime }=1$) with a depolarised mode ($a_{k}^{\prime }/\gamma_{k}^{\prime }=0$), the correction affects their scattering intensity ratio by about 3%. It should be noted that the effect is much larger for spectra recorded with 0° incident laser polarisation, but such spectra will not be discussed in this paper.

Imperfections in the optics, e.g., reduced transmittance for off-axis rays, can be estimated by assuming an uncertainty of 10% for the opening angle, i.e., the limits of integration. However, even such a generous assumption does not yield any noticeable change in the prefactors of ${a_{k}^{\prime }}^{2}$ and ${\gamma_{k}^{\prime }}^{2}$ when recording spectra with 90° incident polarisation (see ESI, Appendix A). Therefore, no effect on the uncertainty of experimental intensities is expected.

#### 2.1.4. Error Due to Transmittance of Monochromator

The sensitivity of the spectrometer is dependent on the state of polarisation of the scattered light. Vertically polarised light (vertical in the lab frame, parallel w.r.t. the scattering plane) is transmitted 1.5–2.5 times less than horizontally polarised light, depending on the wavelength. This factor was determined experimentally; for details, see ESI of Reference [13]. To account for this, it would be necessary to multiply the parallel component of the experimental intensity by said factor. For this, knowledge of the depolarisation ratio of each vibration would be required, which would introduce even more error sources. Our alternative approach is to divide the calculated parallel component $A({\|}^{s})$ by this factor, but to add the resulting uncertainty to the experimental error bar. For simplicity, we apply a homogeneous error of ±0.05 to the transmittance factor, despite its uncertainty being different for different wavelengths. This error will have the biggest influence on a fully depolarised vibration ($a_{k}^{\prime }=0$) in the low wavenumber range, where the factor is ~(1.50 ± 0.05)—its relative contribution to the intensity would then be 1.5%. In order to account for this systematic error, we generously add 1.5% of the experimental intensity as an additional uncertainty to each vibration.

#### 2.1.5. Error Due to Inhomogeneous Illumination

By changing the central wavelength of the monochromator, the spectral window is shifted on the wavenumber axis. Therefore, it is possible for any signal to arrive on the left, middle or right part of the CCD chip, depending on the setting of the monochromator. For an ideal setup, the intensity of a signal should not depend on its position on the CCD. However, this is not the case in our setup. Recording the *Q*-branch of $N_{2}$ and $O_{2}$ from an air expansion and the Rayleigh scattering of residual gas shows that in the lower part (17 of the full range, corresponding to the first 200 pixel columns), there is a drop of intensity by up to ∼15%. This intensity drop is possibly due to the inhomogeneous illumination of the CCD chip. In the remaining 67 of the spectral window, the intensity is rather homogeneous and shows only minor deviations (see ESI, Appendix A).

In this work, quantitative analysis of signals in the lower 17 section of any spectrum is avoided wherever possible. In order to account for the small deviations in the upper 67 section, an error of 1.5% of the intensity is added to the overall uncertainty.

#### 2.1.6. Error Due to Wavelength-Dependent Quantum Efficiency

The quantum efficiency of a CCD camera is dependent on the wavelength of the absorbed light, which has the potential to significantly distort recorded intensities [6]. According to the manufacturer, the CCD camera employed at our setup (Princeton Instruments Pylon40B) shows a rather constant quantum efficiency >95% in the spectral range that was investigated in this work (535 to 580nm). Therefore, we do not assume it has any effect on the uncertainty of experimental intensities.

#### 2.1.7. Error Due to Vibrational Temperature

The scattering intensity of a vibration is influenced by its temperature $T_{v}$ due to the quantum number dependence of the transition moment and the population of higher states. This temperature factor $I_{T}$ for a single harmonic mode [18] is given in Equation (Equation 1), which is based on the assumption that the hot transitions coincide with the cold transitions. The higher the vibrational temperature and the lower the wavenumber, the larger the intensity. For gas-phase spectra, the vibrational temperature is well known, but in supersonic jets, different vibrations cool with different efficiency [19,20].
(1)$$I_{T}\left(\tilde{\nu },T_{v}\right)=\frac{1}{1-\exp(-\frac{hC\tilde{\nu }}{k_{B}T_{v}})}$$

In the previous work of our group, the vibrational temperature was pragmatically assumed at ∼100 K independent of the mode, although low-frequency modes typically cool better than high-frequency modes. Here, we assume that the physically reasonable lowest temperature is 20 K, which is perhaps attainable for very soft vibrations, while the highest value is 180 K. For high-frequency modes, this would be optimistically low, but they do not have a significant excited state population and are temperature-independent in this approximation. For calculated intensities, the average of $I_{T}\left({\tilde{\nu }}_{\mathrm{calc}},20\;K\right)$ and $I_{T}\left({\tilde{\nu }}_{\mathrm{calc}},180\;K\right)$ (with calculated wavenumbers) is used. The relative vibrational temperature error is then estimated as $0.5\times (I_{T}\left({\tilde{\nu }}_{\exp},180\;K\right)-I_{T}\left({\tilde{\nu }}_{\exp},20\;\;K\right))$ (with experimental wavenumbers) and added to the uncertainty of an experimental intensity. Note that this uncertainty becomes completely negligible above ∼600 cm^−1^, relative to other errors discussed in this section.

#### 2.1.8. Combined Error Treatment

In summary, to obtain error bars for the plain experimental signal integrals, the absolute integration error based on statistical noise analysis (see Section 3) is propagated to normalised intensities. Then, the relative error consisting of 1.5% for non-uniform illumination; another 1.5% for uncertainties in the polarisation selectivity of the setup; the temperature error $0.5\times (I_{T}\left({\tilde{\nu }}_{\exp},180\;\;K\right)-I_{T}\left({\tilde{\nu }}_{\exp},20\;K\right))$ using the experimental wavenumber is added. Finally, due to reproducibility and (dimer) impurity issues, 1% of the most intense signal in the covered spectral range is uniformly added to obtain the total uncertainty.

### 2.2. Methyl Methanoate

Building on these preparatory remarks, the study of the simplest ester can form a helpful starting point. In some aspects, such as rotational band contours, methyl methanoate provides a difficult test case. In other aspects, such as conformational diversity, it removes complexity, because, for our purposes, methyl methanoate can be regarded as monoconformational, like the homologous methyl acetate [21].

#### 2.2.1. Comparison to Theory

Figure 4 gives an overview of the fundamental spectrum of methyl methanoate from Raman jet spectroscopy, except for the C−H stretching range. It corresponds to the third-lowest trace of Figure 3 (see Section 3 for experimental details). Due to the finite size of the detector chip, six overlapping spectral windows of 450 to 700 cm^−1^ width [13] were catenated, by matching their intensity for the signals that are marked with vertical bars. This excludes the C−H stretching range in terms of relative intensities, which only offers a limited conformational selectivity only [8]. Better baseline separation compared to the gas-phase spectrum (Figure 1) is evident.

Any comparison with theoretical intensities requires the inclusion of the various experimental corrections outlined in the preceding section. Although it would be desirable to add these corrections on the experimental numbers (as we do for the associated error bars), some of them require theoretical information such as depolarisation ratios; therefore, we make a forward comparison at the level of raw experimental intensities. This also includes the prediction of thermal intensity distortion in the low-frequency experimental data based on the average of the extreme assumptions for the vibrational temperature of 20 K and 180 K. Note that this is done to obtain symmetrical error bars on the experimental values and before normalisation of the theoretical intensities. For convenience, a table illustrating the magnitude of these corrections is given in the ESI (Appendix A), together with a python script that transforms theoretical $a^{\prime }$ and $\gamma^{\prime }$ into intensities comparable to the experimental values.

Experimentally adjusted Raman scattering cross sections were predicted at different levels of theory (for details, see Section 3) and are compared to integrated experimental signal intensities in Table 1 after inclusion of the appropriate experimental corrections. The corresponding calculated wavenumbers are provided in the ESI (Appendix A), where (harmonic) B3LYP/def2-QZVPP shows the best agreement to the (anharmonic) experiment. Note that, for all DFT calculations, D3 dispersion correction was included [22,23], but for brevity, only the functional and basis set are written in the main text. When comparing relative intensities, any normalisation scheme may be applied. Here, we normalise the sum of all fundamentals >500 cm^−1^ that contribute at least 10% to the total scattering to 100. This is done because intense vibrations tend to be predicted better than weak ones and are most useful for conformer quantification, and to exclude the low-wavenumber modes, which have a high level of temperature-induced uncertainty.

Qualitatively, the general trend of intensities is predicted correctly at all levels, but the non-augmented quadruple-ζ basis set predicts a wrong intensity order for $\nu_{8/7/6}$ in comparison to $\nu_{5}$. The usage of basis sets with diffuse functions clearly enhances the description, as is evident from the greatly reduced number of bold-faced entries for the three levels involving diffuse basis sets in Table 1 (>20% deviation representing a rough estimate of typical anharmonic intensity corrections). Because the implementation of the calculation of scattering cross sections in Turbomole was initially reported for PBE0 [24], data for this functional are provided as well. However, no clear advantage of PBE0 over B3LYP is evident.

The largest deviations are found for $\nu_{18}$ and the sum of $\nu_{17/16}$, whose theory-based intensity is overestimated by a factor of two or more within the chosen normalisation method. For these small wavenumbers, the choice of vibrational temperature affects the calculated intensity, and the theoretical overestimation may raise suspicions that the actual vibrational temperature is lower than estimated. However, at ∼100 K the temperature factor for 132 cm^−1^ only enhances the visibility by 26%, for 312 cm^−1^ by a mere 5% in comparison to 0 K. Therefore, even assuming that these vibrations were actually cooled down to 0 K in the expansion would not explain the full deviation. This suggests some deficiency in the electronic structure or harmonic mode description, although one should also reconsider the experimental side and the harmonic thermal correction if other molecules show a similarly pronounced low-frequency gap between theory and experiment.

One possible measure for the deviation between theoretical and experimental intensities can be calculated as follows: In the chosen normalisation scheme, the absolute value of the difference between calculation and experiment reduced by its experimental error, divided by the sum of all calculated fundamental intensities <1800 cm^−1^, gives the relative deviation ϵ from a global spectrum perspective. For the outlier $\nu_{16/17}$, this amounts to 10% for all methods. For the other fundamentals, ϵ is up to 3.5% for B3LYP without and still up to 2% with diffuse functions, respectively. For PBE0 without diffuse functions, ϵ amounts to 4% and with diffuse functions up to 2%. Although these numbers look favourable, due to their reference to the total intensity over the entire fingerprint spectral range and the generous subtraction of experimental errors, they leave room for improvement on the theoretical side. Next to the inclusion of diffuse functions, which helps significantly, going beyond the double-harmonic approximation is a likely improvement. We note the particularly anharmonic character of $\nu_{17}$, which is split into a double peak, and of $\nu_{18}$, the low barrier methyl torsion, whose first overtone shows a large anharmonicity and an intensity comparable to the fundamental. The second overtone of $\nu_{18}$ could be coupled to $\nu_{17}$, possibly provoking the observed splitting through tunnelling. However, we focus on Raman intensities for fundamentals, and therefore will not discuss combination and overtone modes in detail.

In most cases, the intensities of strong fundamentals tend to be described better than weak ones, and the results would suggest the usage of diffuse basis sets. However, diffuse functions increase the computational time significantly and, in combination with finite numerical grids, sometimes cause problems during structure optimisation or frequency analysis. For methyl butanoate, B3LYP/def2-QZVPPD gave small imaginary frequencies for all of its conformers. As a pragmatic approach, the robust non-diffuse method B3LYP/def2-QZVPP is used throughout the rest of this work, but triple zeta basis sets with more or less diffuse functions [25] and Raman-tailored basis sets for the intensity aspect [26] may also work satisfactorily in many cases. Theoretical studies in this direction are encouraged. Ideally, they should not only be compared to higher level calculations, but also to experiments, as in this work.

Note again that theoretical values provided in the tables contain experimental adjustments and corrections based on our setup (for raw data, see the ESI, Appendix A), whereas the uncertainties associated with these corrections are integrated into the experimental error bar. In this way, we still allow for a clean comparison between different (harmonic) theoretical predictions, but, at the same time, indicate the tolerances from the experimental side, when judging the accuracy of theoretical predictions.

#### 2.2.2. Comparison to Gas Phase Reference Data

It would be desirable to have a completely independent Raman intensity measurement for methyl methanoate with a different experimental setup, either in the gas phase or in a supersonic jet expansion. The former is available, albeit with limited experimental details and from the pioneering phase of laser Raman spectroscopy [11]. While we encourage other Raman laboratories to improve upon this situation and to use methyl methanoate as an interesting reference system due to the large theory-experiment gap at low wavenumbers, we will try to form a connection between the pioneering gas phase work and our own. Furthermore, we will compare our own gas phase results with our jet results on a band integral basis, despite the sizeable integration uncertainties due to the extended rovibrational wings. This will shed further light on the advantages (narrow signals and less, but unknown thermal excitation) and disadvantages (aggregation and low molecule density) of the jet approach, before band contour simulations and the move to a conformationally diverse ester are considered.

The previous Raman gas phase study [11] was limited to the brightest fundamentals of methyl methanoate. For example, the $\nu_{18}$ mode was not observed, but correctly predicted from the rotational spectra (131 cm^−1^ [11,27]) and directly observed in infrared studies (130 cm^−1^ [28]). Considering the very low barrier for the associated methyl torsion mode of methyl esters of about 5 kJ mol^−1^ [9], this is not a suitable mode for the benchmarking of harmonic theoretical intensities and thermal effects. Therefore, Table 2 only compares five regions with strong Raman scattering to the gas phase results of the present study (Figure 1) and the corresponding jet results (Figure 4). For a more realistic comparison, intensities of the latter are extrapolated to 298 K by assuming that the vibrational jet temperature can take any value between 20 K and 180 K. This leads to an additional error bar, particularly for low-frequency vibrations, which is added on top of the other error bars. In this way, the problematic extrapolation from an indeterminate jet temperature to room temperature is emphasised for low-frequency modes; however, the room temperature measurement is more problematic in such a case, because the harmonic approximation for the correction factor due to thermal population may be questionable. The theoretical predictions were also extrapolated from 0 K to 298 K. In each case, the total integrated intensity obtained in these five regions was normalised to 100.

We first turn to a comparison of the positions of the signal maxima in the corresponding range, which is appropriate because all intense bands (except for $\nu_{16}$ underlying $\nu_{17}$) are totally symmetric. In our experiment, the wavenumbers of vibrational transitions in the gas phase change very little in comparison to the jet spectra. The literature values [11] differ significantly more, particularly for $\nu_{14}$. The latter may be due to the extensive rovibrational structure (see insert in Figure 1), which is washed out in the low-resolution reference gas phase spectrum, in addition to generic calibration and resolution issues. Clearly, the values of this work are more reliable, and the jet values may be directly compared to theoretical (by preference anharmonic) predictions. Indeed, in all cases excepting $\nu_{14}$, the spread of the explored harmonic predictions does not include the experimental value. It remains to be seen whether this gap between our experiment and harmonic theory can be closed by anharmonic correction or whether intrinsic electronic structure errors dominate.

When comparing integrated intensities, instrumental effects must be considered carefully. Harris et al. used an argon ion laser with excitation wavelengths of 514.5 and 488.0 nm and, although not explicitly stated, we assume that the intensities reported therein were recorded with the 488.0 nm line. In our instrument, the excitation wavelength is 532.27 nm. Additionally, the gas phase data were both assumed to be recorded at 298 K (the actual laboratory temperature in our experiment was 292 K).

Other experimental differences are difficult to correct for. Harris et al. used a Jarrell–Ash Model 500 spectrophotometer, for which technical documentation is unfortunately scarce. The predecessor Model 25–300 is known to collect light at 90° and to count photons; therefore, we assume that this is also true for Model 500 [29]. However, the solid angle of observation is not known. Additionally, Harris et al. used a multipass cell and an optical scrambler in the scattered light beam. No uncertainty for the intensities was reported, but, based on their limited signal-to-noise ratio, a statistical intensity uncertainty of at least 3 in relative intensity units of Table 2 is estimated. In our setup, the laser beam passed the expansion once, no optical scrambler was employed, the cone of observation can be taken into account (see Section 2.1.3) and the dependence of detection sensitivity on the state of polarisation of the scattered photons is known [13]. The uncertainty in these corrections enters the quoted error bar, but their effect is absorbed in the theoretical comparison only. Perhaps the largest uncertainty in the literature intensities is whether they refer to integral or peak intensities. Indeed, the strong deviations for the broad $\nu_{17}$ suggest the latter; therefore, we consider our gas phase values to be more reliable. An independent state-of-the-art reference measurement would be very welcome.

The comparison between the present gas phase and jet data is more productive, until independent data are provided. In view of the effect of thermal excitation and the difficulty in extrapolating rhw very indeterminate vibrational temperature in the jet to room temperature for low-frequency vibrations, the agreement is surprisingly good. Of course, the gas-phase integration suffers from large uncertainties due to overlapping rotational contours, but overall, the thermal effects only seem to be marginally significant for the highest transition ($\nu_{5}$), where the jet and gas phase error bars do not overlap. Comparing the error bars, the jet approach offers error bars that are up to five times smaller, despite the extrapolation in the wrong direction of higher temperature, away from that which theory can best predict.

The overlap between jet and theoretical extrapolations to 298 K is less perfect, as already elaborated in Table 1, but even the gross discrepancy for $\nu_{17/16}$ is now attenuated. The direct comparison of our gas phase experiment and extrapolated theory is fully within the combined error bars, even for the outlier. This shows that jet experiments are needed to challenge theory. We provide a first example; however, this approach has to be repeated for a range of molecules before the improved benchmark potential of jet spectra relative to thermal gas phase measurements can be fully trusted.

In summary, there is overwhelming evidence that our gas-phase spectra are more reliable than the literature ones and that our jet data offer a higher precision for integrated intensities than our gas phase data, although they have to cope with a lower molecular density. We believe that the jet wavenumbers and intensities are also more accurate than the gas phase data, despite the uncertainties associated with aggregation and vibrational temperature. This is largely due to the narrowing of the band profiles under adiabatic expansion conditions and it sets the stage for attempts to simulate this band contour, which will be important for more complex spectra.

### 2.3. Simulation of Rotational Contours

In complex spectra, overlap is unavoidable and an integrated intensity approach, as outlined above, is forced to combine more and more modes into a joint integration. This is not meaningful if more than one species is present, i.e., for conformational quantification (see Figure 2). If one is able to at least semiquantitatively simulate the shape of the Raman band profile, this shape information can be used for deconvolution and fitting of the overlapping spectra. The most simple deconvolution strategy is the analysis of peak maxima and this shall be explored in the following.

Most molecules are asymmetric tops, for which no general analytical description of the rotational states exists. Although energy levels of asymmetric tops may be calculated numerically [30,31] and there are very successful Raman transition intensity treatments for specific cases [32], we are not aware of a universal tool to calculate the latter. However, chain molecules are often close to the symmetric top limit, i.e., Ray’s κ is close to −1 for near-prolate or (less likely for chain molecules) +1 for near-oblate rotors. In such a case, asymptotic energy expressions can be derived and are expected to give reasonable results. If one further assumes that rovibrational coupling such as centrifugal distortion, Coriolis coupling and the vibrational dependence of rotational constants is weak, the same set of rigid rotational constants can be used for the vibrational ground and excited states and the computational cost remains low. If more accuracy is needed, the relevant couplings could also be obtained from vibrational perturbation theory, applied to quantum chemical solutions of the electronic Schrödinger equation [33,34]. Such calculations profit from calibration to experimentally known ground state rotational constants [35], at least for relatively stiff molecules, and the simulation of rotational band contours based on DFT calculations can be automated [36]. However, besides the additional computational cost, we have recently pointed out the limitations of such a VPT2 treatment for low-frequency modes in chain molecules [13]. Therefore, we uses rigid rotor approximation in the present work. In this limit, the total intensity of a vibration is preserved to a very good approximation in the sum over all rovibrational transitions, and we shall use this invariance when introducing approximate band profiles.

In a true symmetric top, the Raman selection rule for ΔJ is 0, ±1 and ±2, and for Δ K either 0, ±1 or ±2, depending on the symmetry of the vibration. In the absence of symmetry ($C_{1}$), no such exclusive selectivity in Δ K applies [37]. Inclusion of all Δ K transitions would artificially increase the total intensity of a vibration in this case. Our ansatz to preserve the sum is to split the anisotropic invariant ${\gamma_{k}^{\prime }}^{2}$ into three parts, which contribute exclusively to either Δ K=0, ±1 or ±2. The separation is based on the values of the components of the transition polarisability tensor $\alpha_{xx},\alpha_{yy}\ldots \alpha_{yz}$. Details are given in the ESI (Section S4).

Lines are convoluted with a Gaussian profile. Its width σ can be adjusted, as can the effective rotational temperature in the case of jet spectra. In Figure 5, the effect of both parameters on the simulated band profiles is exemplified for four modes of methyl methanoate. Despite the pronounced asymmetry of this shortest ester (κ=−0.79), the rovibrational simulation fits reasonably well to the experimental jet spectrum for all employed parameter combinations, when shifted in frequency to correct for anharmonic effects and electronic structure deficiencies. As for the rotational temperature, 10 K is too narrow, but anything between 20 and 40 K gives plausible results. The choice of σ determines the amount of sub-details in the rotational branches and the width of the central $Q^{0}$-branch (ΔJ=Δ K=0). 0.5 cm^−1^ produces too many details in the rotational branches that are not resolved in the experiment, while 1.5 cm^−1^ gives an exaggerated smoothing, as long as hot band and rovibrational contributions (which the rigid rotor approach cannot capture) are absent and the theoretical calculation predicts the correct anisotropy of the polarisability tensor (see ESI, Appendix A). Based on these findings, 30 K and 1.0 cm^−1^ will be used for the simulation of jet spectra.

Figure 6 shows the limits of the rigid rotor approach when trying to simulate the 298 K gas phase spectrum. The $Q^{0}$-branches stay sharp, because the excited state rotational levels match the ground state levels, but the rotational structure near the baseline is still reproduced reasonably well for the fundamental transitions, considering the approximations involved. One could empirically vary the only remaining fitting parameter σ (besides the wavenumber scaling) to account for each individual observed band profile, but we prefer to focus on the low-resolution jet spectra, where the rigid rotor approximation is more realistic. Considering that methyl methanoate is close to a worst-case scenario in terms of its size and asymmetry in the family of chain esters, we expect a satisfactory performance regarding our goal of disentangling longer-chain ester conformations.

### 2.4. Rovibrational Simulation of Methyl Methanoate

As outlined above, the goal of the rovibrational simulation is to match the experimental peak heights. How well these fit for all vibrations <1800 cm^−1^ of methyl methanoate is shown in Figure 7. If the simulated peak heights matched perfectly, there would be no need to individually scale the intensity. However, it must be stressed that the calculated peak heights cannot be expected to fit the experimental spectrum better than the underlying scattering cross sections. Therefore, the direct comparison of these factors to the reference ratio of integrated experimental intensity to calculated cross section from Table 1 is provided in Table 3.

The deviation between simulated peak height and experiment follows the same trend as the calculated cross section, but no clear advantage is visible for any of the widths. Modes with sharp contours (i.e., with a strong $Q^{0}$-branch) are described better than modes with broad contours. In some cases, such as $\nu_{9}$, the peak height deviates less than the cross section, probably due to fortunate error compensation. In the case of $\nu_{17}$ the experimental peak height is approximately halved due to tunnelling splitting, which the harmonic simulation cannot reproduce. A global choice for an optimal simulation width is not straightforward, because none of them seem to be optimal. For some modes, such as $\nu_{18}$, the broad simulation (1.5 cm^−1^) gives the best result and the narrow simulation (0.5 cm^−1^) deviates very strongly, while for others, such as $\nu_{8/7/6}$, the opposite is observed. In cases, such as $\nu_{13}$, only the middle case (1.0 cm^−1^) agrees with the reference. Any of the choices based on contour simulations is better than simply taking the calculated Raman cross section as a measure for experimental peak heights ($P_{h}$), because of the wide variations in rotational contour.

Overall, a σ of 1.0 cm^−1^ is considered the best compromise, yielding the fewest extreme outliers. It must be noted that the error introduced by the arbitrary choice of this width is smaller than the general deviation between calculated harmonic and integrated experimental Raman intensities. As a result, it can be stated that the rovibrational simulation does not introduce grossly misleading errors and can, therefore, be considered as a useful tool for quantitative analysis, particularly if it is based on several vibrations of the same species.

A further illustration of the comparability of integral and peak height approaches to the quantification of Raman intensities is shown in Figure 8, which correlates different fractional experimental and theoretical transition intensities in the spectrum. By plotting the fraction of the total intensity in the full spectral window, deviations in strong signals are emphasised. Such strong signals are particularly useful for conformational quantification if they are sufficiently separated from other signals. The plot also correlates the sum of the two valence vibrations ($\nu_{14}+\nu_{5}$), where the performance of integral and peak height methods is quite similar. The relatively large deviation for $\nu_{5}$ is mostly cured by including diffuse functions (see the ESI, Appendix A)

Based on the validation of the peak intensity simulation for Raman spectra of methyl methanoate, this tool will now be applied to the partial conformational relaxation of methyl butanoate.

### 2.5. Methyl Butanoate

#### 2.5.1. Quantum Chemical Predictions

Methyl butanoate has three single bonds, which support several conformations. The C−O−C−C_*α*_ torsional angle can be either *trans* (180°) or *cis* (0°), while the other two, O−C−C_*α*_−C_*β*_ and C−C_*α*_−C_*β*_−C_*γ*_, can be *trans* (180°) or *gauche* (±60°), giving a total of 18 possible conformations. However, most of them can be disregarded. The *cis* ester conformation adds an energy penalty of ∼30 kJ mol^−1^ on B3LYP/def2-QZVPP level; therefore, only the *trans* C−O−C−C_*α*_ conformations need to be considered here [38], in analogy to carboxylic acids [39]. Additionally, in classical Raman spectroscopy, enantiomers can not be distinguished; therefore, only one member of the enantiomeric pairs will be considered. Lastly, the conformer with two heterochiral *gauche* conformations for the C−C bonds is not a minimum structure at B3LYP/def2-QZVPP level. Hence, there are only four relevant conformers, as follows: The first letter denotes the conformation of C−O−C−C_*α*_ and is always t (*trans*), the second and third letter denote the conformation of O−C−C_*α*_−C_*β*_ and C−C_*α*_−C_*β*_−C_*γ*_, respectively, and can be either t (*trans*) or g (*gauche*). Note that some angles deviate quite strongly from the ideal values of ±60° and ±180°, but we refrain from using a more differentiated nomenclature [40].

A complete torsional scan of the B2PLYP-D3(BJ)/aug-cc-pVTZ PES has already been performed [9]. In this work, only the stationary points were recalculated at the CCSD(F12*)(T*)//B3LYP level. Pictures of the conformers, their relative energies and barriers of interconversion are shown in Figure 9 and in Table 4, including a pioneering DFT study [6]. The implications for the experiment shall be briefly discussed here.

At room temperature (i.e., prior to expansion), the population is Boltzmann distributed. Using the differences in Gibbs energies at 298.15 K (electronic energy from CCSD(T) and thermal enthalpy and entropy from B3LYP) and the twofold statistical advantage of the chiral enantiomeric pairs a population of ttt = 27%, ttg = 34%, tgt = 24%, tgg = 15% is found. These populations differ by less than four percentage points if the CCSD(T) correction is removed.

Downstream a supersonic expansion, the molecules are collisionally cooled and, therefore, try to relax into the global minimum, for which the barriers of interconversion need to be overcome. A typical estimate for our setup is that barriers <5 kJ mol^−1^ are easily overcome, whereas barriers >10 kJ mol^−1^ are largely insurmountable.

Our values suggest that the tgt minimum is higher than the barrier to ttt, once the ZPE (without imaginary contribution at the barrier) is added. Therefore, it is expected that, in a supersonic jet, it will fully interconvert into ttt. However, a small adiabatic barrier should not be ruled out [9] and would still be consistent with easy relaxation. ttt and ttg will not interconvert significantly and will both be present, which can be described as conformational freezing, but is also supported by their very similar energy. No clear expectation for the probability of interconversion of tgg into ttg can be derived from the relative energies.

The effect of the ester group is very prominent: the adjacent C−C_*α*_ bond is much more flexible than the purely alkyl one. This effect is nullified if the alkanoate part of the ester is branched at the α position, which was shown in a recent microwave study on ethyl 2-methylpentanoate [41]. There, the observed minimum structure has a O−C−C_*α*_−C_*β*_ angle of 143.1°, which can be described as a slightly tilted ttt structure in our nomenclature. The barrier towards a conformer similar to tgt was predicted to be around 10 kJ mol^−1^ on various levels of theory. Additionally, the potential well around ttt was predicted to be rather flat and the O−C−C_*α*_−C_*β*_ angle of the minimum structure differed by ±20°, depending on the method employed. At the MN15 hybrid density functional level, a double minimum was predicted, contradicting the experimental results.

#### 2.5.2. Experimental Spectra

The ν(C=O) fundamental (1750 to 1800 cm^−1^) proved to be a useful clustering marker for methyl methanoate, and, therefore, an analogous variation in the ester concentration was found for methyl butanoate (Figure 10). The signals at 1742 and 1753 cm^−1^ show the most prominent changes w.r.t. ester concentration and can, therefore, be attributed to dimers and larger clusters on top of monomer signals. Monomeric ν(C=O) shows remarkable splittings that might at first be attributed to the different conformers. However, the experimental splitting (1760, 1764, 1768 and 1772 cm^−1^) is much larger than the harmonically predicted one (1785, 1786, 1787 and 1788 cm^−1^); therefore, a splitting due to anharmonic coupling with combination modes seems more likely. It seems that such a splitting partially survives in the crystalline state [6]. Interestingly, the position of the ν(C=O) fundamental is a sensitive indicator of fatty acid chain length in the liquid state [42], despite this gas phase complexity. A similar concentration series in the low-wavenumber region is provided in the ESI (Appendix A). As no difference is observed between the lowest two traces, the settings with a relative concentration of 1.5 were used for further analysis.

By varying the expansion conditions, spectra with differently pronounced collisional cooling can be obtained. In Figure 11, the low-wavenumber range of methyl butanoate recorded in the gas phase as well as in five jet expansions with different conditions is shown alongside selected intense and separated lines of all four conformers taken from quantum chemical calculations. This spectral range was chosen because it has a pronounced conformational discrimination potential. In the room-temperature gas-phase spectrum (top trace), only some of the signals can be assigned in a straightforward way to conformers, such as 335 and 433 cm^−1^ to ttt and 371 and 463 cm^−1^ to ttg, but no reliable quantitative information can be obtained because of the overlap of the rotational branches. This underscores the need for supersonic expansions. In the hottest expansion (a) the rotational structure is much more narrow, and despite a nozzle temperature of 150 ${}^{\circ }C$, the effective rotational temperature in this spectrum is substantially lower than room temperature. The two vanishing signals near 638 and 752 cm^−1^ can be tentatively assigned to tgt, suggesting that even mild cooling effects completely funnel the initial tgt population to ttt.

If the expansion is probed further away from the nozzle, the collisional cooling process is more complete: The hot band structure at 463 cm^−1^ is reduced from (a) to (b). If the nozzle is not heated during the expansion, even stronger cooling can be achieved (traces c and d). The partial relaxation of tgg into ttg can be tracked in the region around 600 cm^−1^: the intensity of the tgg signal at 604 cm^−1^ relative to the ttg signal at 585 cm^−1^ decreases from (a) to (d), but does not vanish, even at larger nozzle distance (not shown). By substituting 20% helium carrier gas by argon (e), the cooling efficiency is increased due to the improved mass matching or transient attachment [43]. Here, tgg is completely depleted and only the two most stable conformers ttt and ttg remain.

With 15% argon (not shown), a trace of tgg is still observable, while larger amounts of argon promote dimer formation. Trace (e) thus represents the simplest spectrum. The small surviving signal marked o? is possibly the (unusually strong) overtone of the ttt COC bending vibration near 310 cm^−1^, but other combinations, a ttg origin, or strongly Raman-active impurities cannot be ruled out. In the OH stretching range (3400 to 3850 cm^−1^), no other signals were observed, except for a small water signal at 3656 cm^−1^; therefore, impurities due to the direct dissociation products of methyl butanoate, methanol or butanoic acid, are not expected to contribute significantly to the low-frequency spectrum. This qualitative analysis shows how low barriers of interconversion are overcome in a supersonic expansion, and how high barriers lead to conformational freezing.

The coldest spectrum (Figure 11e) will now be used to quantify the ratio of ttt and ttg. In Figure 12 a section of the jet spectrum is shown along different simulated traces for these conformers. For comparison, the left part is the same as in Figure 2. In the middle part the rovibrational simulation of ttt and ttg, as well as the sum of both (red), is shown. As was performed for methyl methanoate in Figure 7, the simulated peak heights have to be multiplied by the factors shown above the corresponding signals in order to match the experimental spectrum. This can be done for the two main signals, but the harmonic calculation predicts that the two signals at ∼885 cm^−1^ are too close to each other, so that their sum can not be properly fitted to the experiment. An improved simulation can be obtained if, instead of the harmonically calculated wavenumbers, the experimentally observed ones are used for the central wavenumber of the rovibrational simulation. This is shown in the right part. Here, the simulation becomes almost indistinguishable from the experimental spectrum, and the peak height factors for both ttg signals at 868 and 881 cm^−1^ can be determined.

Therefore, a two-step analysis of the full spectrum is employed: First, the rovibrational simulation with harmonically predicted wavenumbers is used to assign all relevant signals in the experimental spectrum, then the simulation is redone using the experimental wavenumbers, and finally the peak heights for each assigned signal are fitted. The results are shown in Figure 13. We note that the ttt signal at 338 cm^−1^ is at the edge of the poorly illuminated part of the CCD (vide supra) and, because of that, its experimental intensity might be slightly underestimated. This datapoint is still included in the analysis, because it does not seem to be strongly distorted. From these 7 (ttg) +4 (ttt) numbers the mean value and the population standard deviation (1N∑(x−x¯)2)) are formed: ttg = 1.09 ± 0.39, ttt = 0.92 ± 0.09. These are normalised to give 100 in sum, and the uncertainty is propagated, giving: ttg = 54 ± 9, ttt = 46 ± 9.

For comparison, those 5 + 3 signals that were spectrally separated enough were integrated, and the integrals were divided by calculated Raman cross sections, which are shown as bold numbers in Figure 13. The individual errors were determined similar to methyl methanoate (vide supra). The integration error was estimated from statistical noise analysis (see Section 3) and propagated in the normalisation step to the ttg signal at 466 cm^−1^. Then, the relative temperature error and 3% of the intensity due to non-uniform illumination and uncertainty in the polarisation selectivity were added, and finally 1% of the most intense signal (ttg at 868 cm^−1^) was added uniformly, due to reproducibility and (dimer) impurity issues. Again, from the mean and population standard deviation (excluding the ttg signal at 868 cm^−1^), the final ratio for ttg:ttt of 57:43 is obtained. The error bar is ±5 without and ±8 with inclusion of the individual integration errors.

Both of our methods for the determination of conformational ratios agree with each other, and, as expected, the peak height method provides a higher level of uncertainty than the integral method. This ratio has also been determined by microwave spectroscopy in supersonic expansions, where a similar conformational freezing was observed and a ratio of ttg:ttt of (59 ± 6):(41 ± 4) was found [9]. Both of our results agree with the reference values. Hence, our Raman setup is capable of distinguishing different ester conformers and semi-quantifying their abundance.

With a similar procedure, the peak heights of the coldest, argon-free spectrum from Figure 11 (trace d) were analysed as well, yielding the relative abundance: ttg = 53 ± 9, ttt = 45 ± 9, tgg = 2.0 ± 0.4.

The main conformers ttt and ttg were previously identified by a combination of quantum chemical calculations and Raman spectroscopy of solid methyl butanoate [6]. There, a strong connection between basis set size and quality of the theoretical prediction was observed. Already the gas phase B3LYP/6-31G(d) approach, which is superseded by far more accurate methods at present, was sufficient for a reliable signal assignment based on the vibrational wavenumbers, despite the difference in phase state. However, the corresponding Raman intensities were insufficient for conformer quantification. Reasonable intensities that qualitatively matched the experimental spectrum were obtained with the larger Sadlej pVTZ basis set [44]. Our calculations with the even larger def2-QZVPP basis set are also in reasonable agreement for most of the fundamentals, but quantitative matching is only obtained if a favourable anharmonicity effect of up to 20% is assumed (see ESI, Appendix A).

Table 5 compares the experimental isomer ratios with theoretical models which all assume that the torsional angle around the C−C*_α_* bond relaxes completely to its global minimum (t) value, consistent with the experimental observation under cold jet conditions. The outer torsional angle distribution is assumed to either freeze at the nozzle (300 K) or to relax to a conformational temperature of 200 or 100 K. 100 K is not very likely given the height of the barrier, but is still included for comparison. One might further assume that the rotational and vibrational partition functions are the same in both surviving conformations (simple Boltzmann model, $\Delta E_{0}$) or include those partition functions explicitly into the population calculation before relaxation (entropy-inclusive $\Delta G^{Ꝋ}$ model in the harmonic/rigid rotor approximation). In Reference [9], $\Delta E_{0}$ values are actually used in the model, whereas $\Delta G^{Ꝋ}$ values were intended [45]. We also tabulated the $\Delta G^{Ꝋ}$ values obtained from the authors [45], for convenience. Both models based on MP2 energies disagree with all three experimental estimates. The $\Delta G^{Ꝋ}\left(300\;K\right)$ model based on CCSD(T)-edited B3LYP calculations gives the same prediction as the one based on B2PLYP energies, and it still disagrees with two experimental evaluations. The other models agree with two or three experimental values within their error bars, and we cannot decide between them.

In summary, the separation of the C_*α*_−C_*β*_ torsional families (tgt, ttt) and (tgg, ttg) in the jet expansion is either complete (no transfer between them, corresponding to $\Delta E_{0}\left(300K\right)$, neglecting partition function differences) or partial ($\Delta G^{Ꝋ}$(<300 K), with some C_*α*_−C_*β*_ torsional relaxation despite the significant barrier and small energetical driving force).

## 3. Materials and Methods

### 3.1. Quantum Chemical Calculations

Optimised structures, harmonic frequencies and Raman intensities were obtained for the B3LYP and PBE0 density functionals and def2-QZVPP, def2-QZVPPD and aug-cc-pVQZ basis sets using Turbomole [46]. Unless stated otherwise, results are shown for B3LYP/def2-QZVPP. In all cases, Grimme’s dispersion correction D3 with Becke–Johnson damping (BJ) and three-body terms (abc) was applied [22,23], and for speedup multipole accelerated resolution of identity for the Coulomb terms MARI-J was used [47]. A frequency analysis was performed for the main isotopes of H, C and O, using $C_{s}$ symmetry were applicable. Calculated wavenumbers are unscaled unless stated otherwise. Transition states were checked to show a single imaginary frequency. An SCF convergence threshold of 1 × 10^−9^
*E*_h_, an energy change threshold during structure optimisation of 1 × 10^−6^
*E*_h_ and the integration grid m5 were used. For all other parameters the default values were used.

Single point energies were calculated on RIJK-CCSD(F12*)(T*)/cc-pVTZ-F12 level [48,49,50,51] (SCF convergence 1 × 10^−9^
*E*_h_, frozen core) using the B3LYP/def2-QZVPP optimised structures and combined with the B3LYP zero-point energy. Relative Gibbs energies at various temperatures were calculated using the B3LYP enthalpy and entropy and the CCSD(T) electronic energy.

For reference, CCSD(T) single-point energies were also calculated for structures that were optimised on DF-CCSD(T)-F12A/cc-pVDZ-F12 level (see ESI, Appendix A). From those, the error of relative energies of minima due to structure uncertainty introduced by optimising on B3LYP is estimated at <0.1 kJ mol^−1^.

### 3.2. Rovibrational Simulation

Rotational broadening of vibrational transitions was simulated by explicitly calculating the positions and intensities of rovibrational lines, for details about the underlying theory see Section 2.3 and ESI (Section S4). Typically, for each molecule, up to a few million lines were simulated and subsequently folded by Gaussians on a 0.1 cm^−1^ grid. For speedup, line positions are rounded with 0.01 cm^−1^ precision prior to folding, and the calculation of Gaussians is limited to ±5σ. The script is written for GNU Octave [52]. The simulation is typically finished within a minute on a single core in the current implementation.

The data needed for the simulation are the vibrational wavenumbers, the rotational constants and the components of the transition polarisability, which can be obtained from any common Raman calculation. A detailed explanation of how to extract these components from a Turbomole output is provided in the ESI (Section S5.2).

### 3.3. Experimental Setup

Raman spectra were recorded in a continuous supersonic expansion; the details of the setup are described in References [10,13]. In short, the ester is strongly diluted in helium (typically <1%) and is continuously expanded at 0.7 bar through a heatable 4 nm × 0.15 mm slit nozzle into a vacuum chamber at ∼1 mbar. Mass flow controllers in the gas feeding line allow for the mixing of up to three different gases in a controlled manner. Parallel to the nozzle, the expansion is probed by a mildly focused 532 nm laser beam (Spectra Physics Millennia 25 eV) perpendicular to the expansion. By moving the nozzle relative to the laser beam, different regions of the expansion can be probed. In this work, the distance was varied between 0.5 and 1.0 mm. For gas phase spectra, the vacuum pumps were shut off and the chamber was filled with the gas mixture at a fraction of 1 bar.

The scattered photons are collimated at 90° with a camera objective, led out of the vacuum chamber, dispersed in a monochromator and detected on a CCD camera (Pylon 40 B). Photons were accumulated for up to 10 min per scan; spikes due to background radiation were removed by comparing several scans in an automated way.

Spectra were calibrated using atomic Ne emission lines [53], the readout offset of the CCD (see ESI, Appendix A) was subtracted and, after additional baseline correction, the intensities were corrected for the nonlinearity of the wavenumber axis by dividing each pixel intensity by its spectral width.

### 3.4. Integration of Signals

Experimental intensities were determined by integration using a tool called NoisySignalIntegration.jl, written in Julia [54,55]. In short, the uncertainties of integrals were simulated based on two effects. Random noise was added to the spectrum, and the lower and upper integration border were varied randomly within some user-specified bounds. For each signal, 10,000 random samples were integrated and a linear baseline was subtracted. From these random draws, the mean and standard deviation were determined and used as the signal intensity.

Signals that were not sufficiently separated from neighbouring signals (e.g., in the gas phase) were integrated without baseline correction and random noise sampling; instead, the value of a linear baseline correction was taken as the uncertainty.

### 3.5. Chemicals

Methyl methanoate ≥99.0% from Fluka, methyl butanoate ≥99.0% from TCI, helium 4.6 from Nippon Gases, and argon 5.0 from Linde AG were used as supplied.

## 4. Conclusions

The conformational quantification of chain molecules in the gas phase typically requires spectroscopic tools, which are sensitive to different conformations and allow for an accurate quantum-chemical calculation of the underlying spectral transition moments. No single technique is perfect, and it is important to use different approaches to ensure mutual consistency. This work proposes linear Raman jet spectroscopy, besides the more widespread microwave rotational studies and IR/UV double- and single-resonance techniques for this goal. It provides combined access to the room-temperature gas-phase and adiabatically cooled samples for spectral simplification and conformational relaxation. In the liquid phase, many conformational details are washed out [42] and the connection to theoretical predictions is much more demanding.

Using the conformationally uniform methyl methanoate, we identified possible error sources and uncertainties in relating computed Raman scattering cross sections to experimental signal strengths. This includes instrumental parameters, clustering indicators, hot band contributions, the impact of diffuse basis sets and the double-harmonic approximation. Particularly for methyl torsional motion, the latter was found to be far from accurate. As larger and conformationally diverse molecules suffer from spectral signal overlap, we developed a band shape simulation procedure of sufficient accuracy for near prolate (and, less relevant for chain molecules, near-oblate) tops to allow for a simple quantification via sharp *Q*-branches and to open the way for contour-fitting strategies.

These insights and tools were then applied to methyl butanoate, a highly flexible ester which serves as an entry point for large amplitude chain-folding studies in the gas phase driven by London dispersion. The latter was previously investigated for linear alkanes [4], where Raman spectroscopy remains the only available spectroscopic tool, and to alkylbenzenes [8], where UV and UV/IR spectroscopy provide semiquantitative access to the conformational distribution and the aromatic ring helps in the folding process. For methyl butanoate, we were able to confirm that suitable jet cooling reduces the number of distinguishable chain conformers of a hexane chain (10) in a simplified rotational isomeric state model to only 2 (ttt, ttg), if a if a −CH_2_−CH_2_− segment is replaced by an ester group −O−C(=O)−. The only surviving isomerism in the ester refers to the alkyl chain on the acid end. The complexity reduction results from a fixation on the alcohol end (10→5) and from a softening of the conformational landscape adjacent to the acid group (5→2). We were also able to confirm that the remaining, nearly isoenergetic isomerism can be viewed as largely, but perhaps not completely, frozen with respect to the terminal CCCC torsional angle. Furthermore, we identified sufficiently accurate quantum chemical approaches for the prediction of Raman intensities, which are scalable to longer chains. These findings are promising for our goal to use mid-chain esters for the acceleration of large amplitude stretched-to-folded global minimum transitions in chain molecules as a function of chain length. With the quantification tools developed in this work, it should be possible to identify the onset of folded isomer population, despite their expected conformational and spectral complexity.

## Figures and Tables

**Figure 1 molecules-26-04523-f001:**
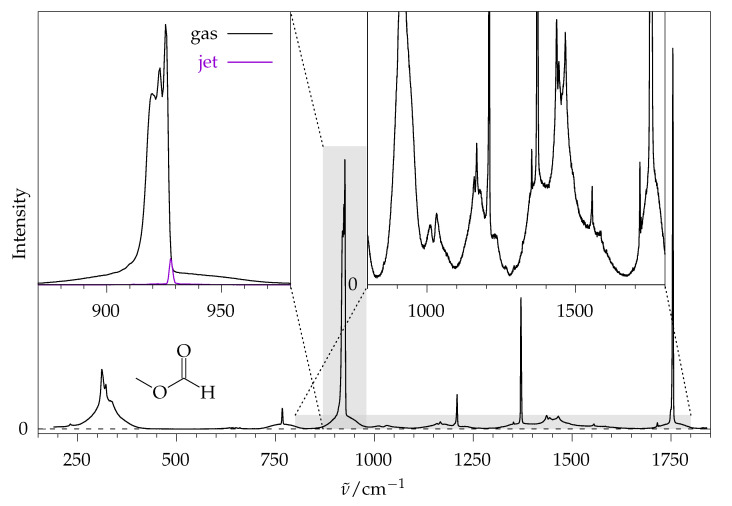
Gas-phase Raman spectrum of methyl methanoate. Left inset: Enlarged part of the ester linkage stretching mode $\nu_{14}$ (vide infra) in comparison to a jet cooled spectrum scaled to be part of the gas phase spectrum. Right inset: Enlarged view of the baseline, which does not drop to 0 due to thermal excitation.

**Figure 2 molecules-26-04523-f002:**
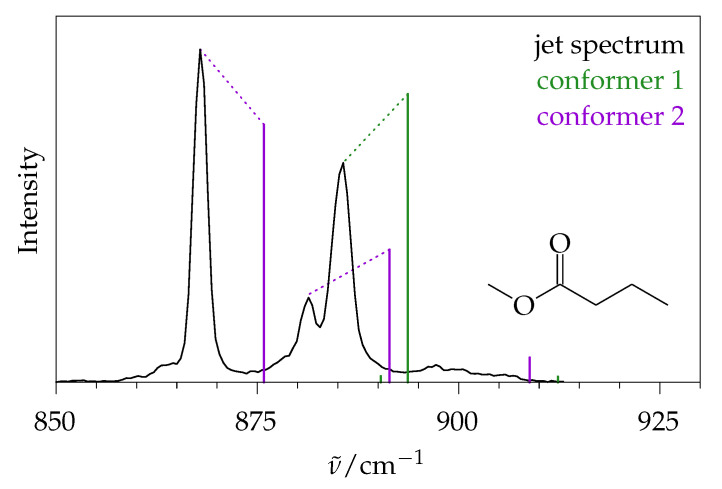
Enlarged part of a Raman jet spectrum of methyl butanoate (line) and calculated harmonic Raman spectrum for its two main conformers (vertical bars encoding predicted wavenumber and integrated intensity). Although a qualitative assignment is directly possible by implying a systematic anharmonic downshift, a quantification by signal integration is not, because the baseline does not drop to zero between the signals.

**Figure 3 molecules-26-04523-f003:**
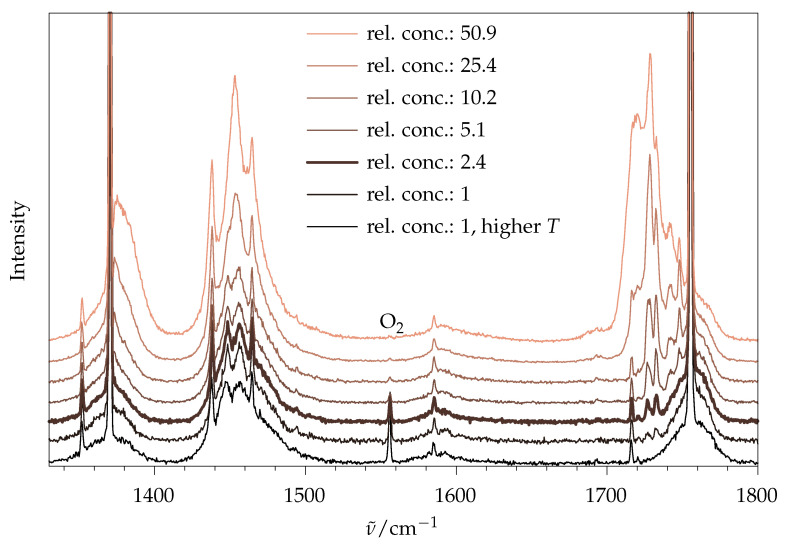
Raman jet spectra of methyl methanoate for widely different ester concentrations, scaled to the peak monomer signal at 1755 cm^−1^. The expansion conditions are identical, except for the lowest trace, where a reduced distance to the nozzle (0.5 mm) leads to higher temperature. The $O_{2}$ signal is due to air impurity. The relative concentration of 2.4 was chosen for further analysis (bold trace).

**Figure 4 molecules-26-04523-f004:**
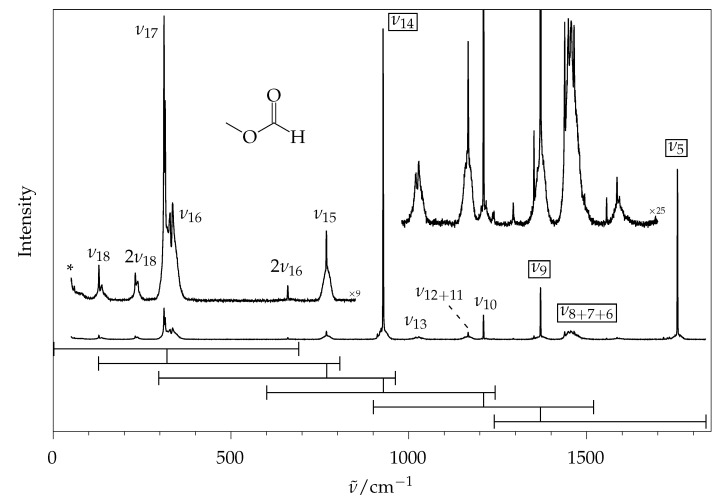
Experimental jet spectrum of methyl methanoate, constructed from six overlapping segments (horizontal bars, intensity matching at the vertical connection lines). * marks air impurities, any signal <50 cm^−1^ is distorted by a Rayleigh edge filter. The band labels $\nu_{5}-\nu_{18}$ ignore symmetry, see Table 1 for the associated dominant vibrational character and symmetry. Signals that are used for normalisation in Table 1 are framed.

**Figure 5 molecules-26-04523-f005:**
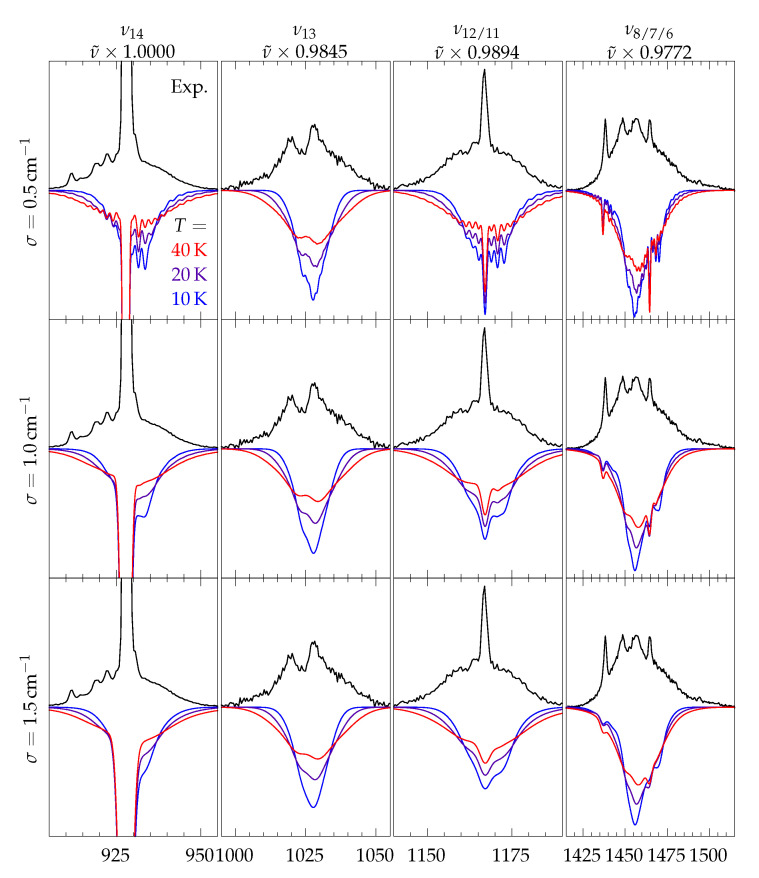
Near-baseline sections of the jet spectrum of methyl methanoate (black trace) and rovibrational simulations for different temperatures *T* and Gaussian widths σ. Calculated wavenumbers are scaled by the factors provided below the vibrational labels and calculated intensities are scaled to match the integrated experimental intensity.

**Figure 6 molecules-26-04523-f006:**
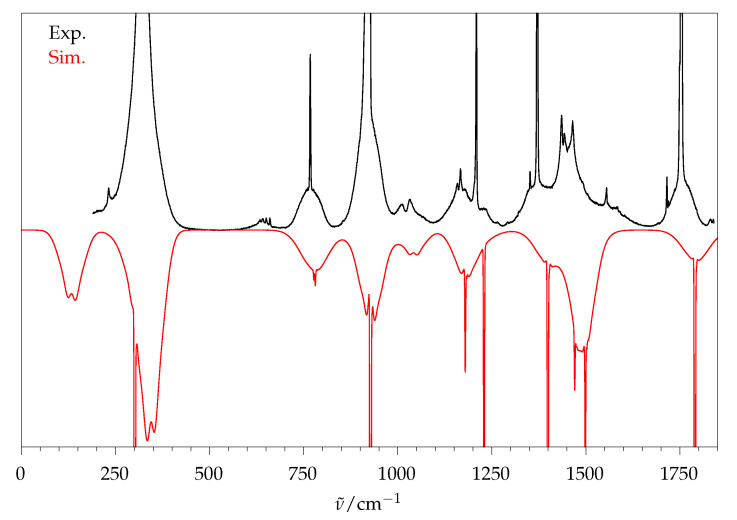
Experimental gas phase spectrum of methyl methanoate (**top black trace**) and rovibrational simulation for 298 K and σ=1.0 cm^−1^ (**bottom red trace**).

**Figure 7 molecules-26-04523-f007:**
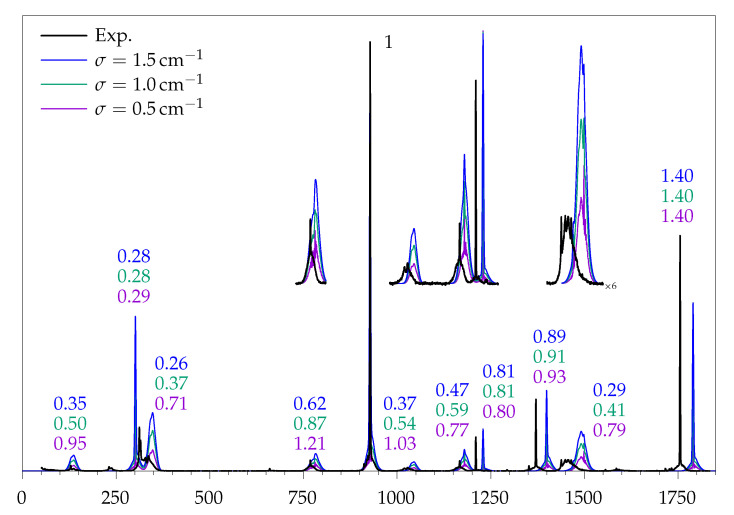
Black trace: Experimental jet spectrum of methyl methanoate. Coloured traces: rovibrational simulations for different Gaussian widths, matched to the strongest peak at 928 cm^−1^. In order for the calculated peak height of a vibration to match the experiment, the intensity of the simulated spectrum has to be multiplied with the numbers given.

**Figure 8 molecules-26-04523-f008:**
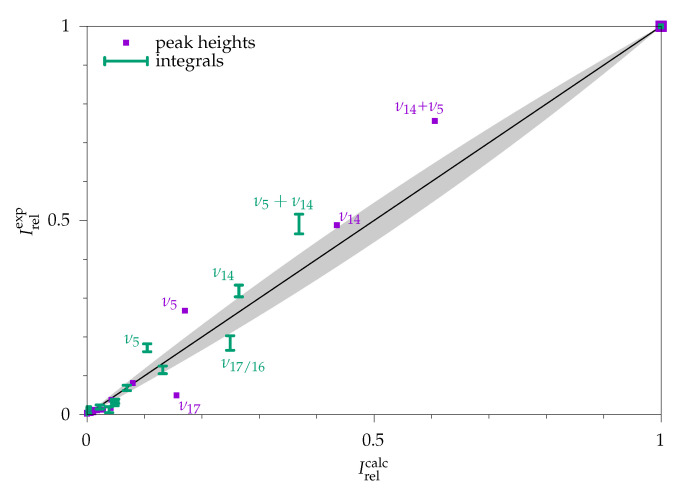
Ratio of calculated and experimental intensities for methyl methanoate. Violet points: peak height method (σ=1.0 cm^−1^). Green error bars: integral method; both normalised to give 1 in sum. Selected strong fundamentals and sums of fundamentals are labeled to exemplify that the deviations between experiment and theory are often correlated among the two methods. The area that would be covered by an anharmonicity effect of up to 20% is shaded in grey.

**Figure 9 molecules-26-04523-f009:**
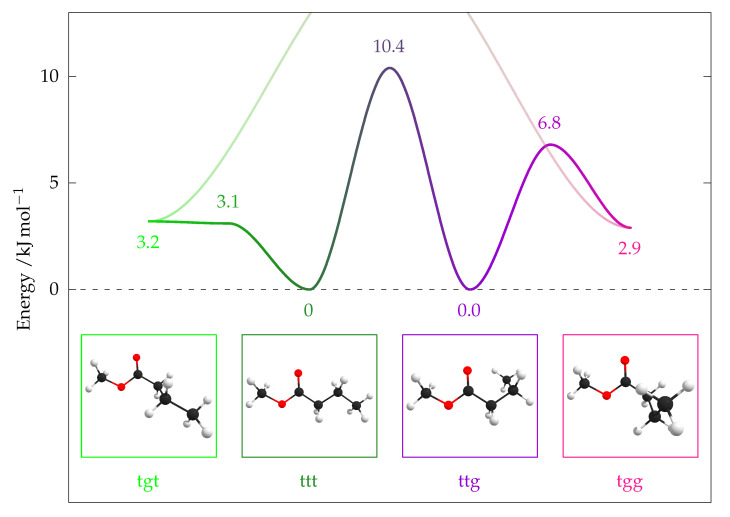
Relative energies and barriers of interconversion of the main conformers of methyl butanoate on CCSD(F12*)(T*)//B3LYP level (ZPE included). Isomerisation about the C${}_{\alpha }$C${}_{\beta }$ bond is more feasible if the OCC${}_{\alpha }$C${}_{\beta }$ dihedral is in t orientation (the barrier for g orientation exceeds the displayed energy range).

**Figure 10 molecules-26-04523-f010:**
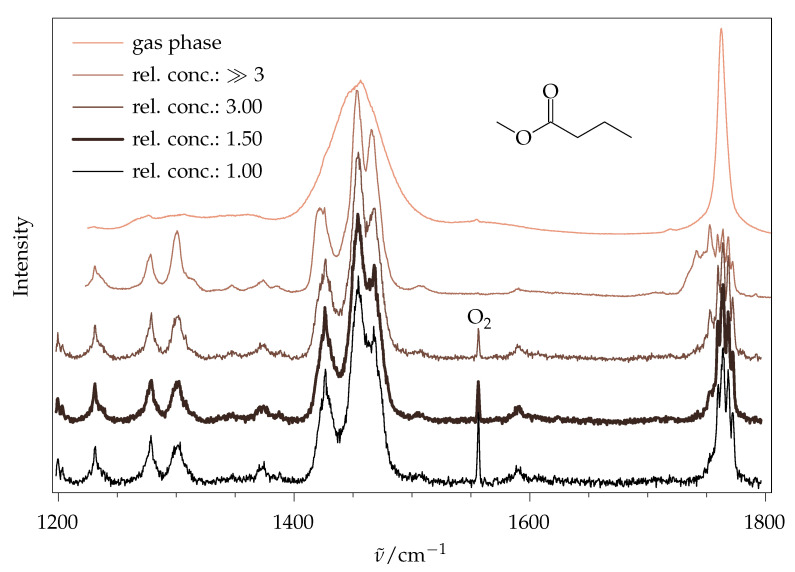
Raman spectra of methyl butanoate. Top trace: gas phase. Lower four traces: jet spectra with identical conditions, except for varying ester concentration. $O_{2}$ and $H_{2}O$ signals at 1556 and 1590 cm^−1^ are from air impurity.

**Figure 11 molecules-26-04523-f011:**
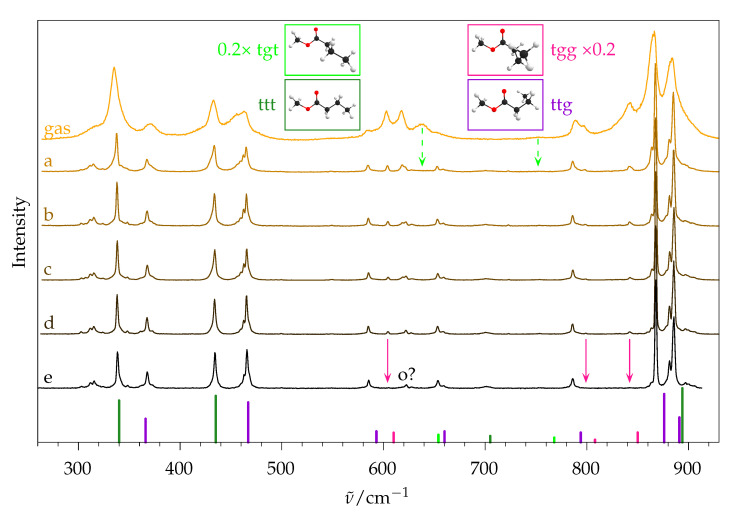
Raman spectra of methyl butanoate, sorted by descending effective temperature. Top trace: gas phase spectrum at room temperature. a–e: jet spectra with same substance concentration (1.50 in Figure 10), but different nozzle temperatures $T_{N}$ and distances between nozzle and laser beam *r*. (**a**) *T*_N_ = 150 °C, *r* = 0.5 mm. (**b**) *T*_N_ = 150 °C, *r* = 1.0 mm. (**c**) *T*_N_ = room temperature, *r* = 0.5 mm. (**d**) *T*_N_ = room temperature, *r* = 1.0 mm. (**e**) like d with 20% helium replaced by argon. Vertical bars at the bottom: selection of intense lines of the four conformers, relative intensities of tgt and tgg scaled by 0.2.

**Figure 12 molecules-26-04523-f012:**
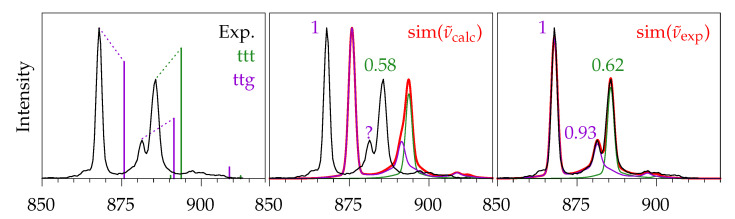
Jet spectrum of methyl butanoate (black, Figure 11, trace (e)) and three differently simulated spectra. **Left**: Raman cross sections indicated by vertical bars like in Figure 2. **Middle**: Rovibrational simulation, using calculated harmonic vibrational wavenumbers. The question mark indicates the inability to quantify the weaker ttg signal due to overlapping simulated band contours. **Right**: Rovibrational simulation, based on experimental vibrational wavenumbers.

**Figure 13 molecules-26-04523-f013:**
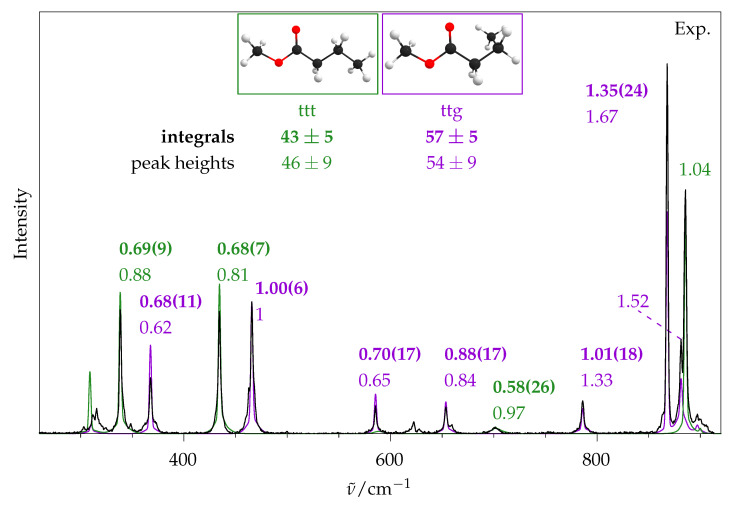
Determination of conformational ratio for methyl butanoate. Black trace: argon-enriched jet spectrum (Figure 11, trace (e)). Coloured traces: rovibrational simulations of ttt and ttg, using experimental wavenumbers (except for the signals around 300 cm^−1^, which are not included in the analysis anyway). Non-bold numbers without uncertainty: factors, by which the simulated peak height has to be multiplied to match the experiment. Bold numbers with uncertainty: Ratio of integrated signal intensity and calculated scattering cross sections. Normalised to the ttg signal at 466 cm^−1^.

**Table 1 molecules-26-04523-t001:** Experimental wavenumbers $\tilde{\nu }$ and relative intensities $I_{e}$ (normalised to $\nu_{5}+\nu_{6}+\nu_{7}+\nu_{8}+\nu_{9}+\nu_{14}=100$) of methyl methanoate modes $\nu_{n}\left(\Gamma \right)$ compared to intensities calculated on different levels in the double-harmonic approximation: B = B3LYP, P = PBE0, d = def2-QZVPP, dD = def2-QZVPPD, a = aug-cc-pVQZ, in bold, if the harmonic value disagrees with (anharmonic) experiment in its error bars by more than 20%. For vibrations without a sharp *Q*-branch, the average of the wavenumbers of the rotational maxima is given. Description of vibrations: ν stretching, δ bending, τ torsion, ω wagging, ip/op in plane/out of plane, (a)s (a)symmetric. ^*a*^ An asymmetric double peak is observed.

Mode	ν˜/cm−1		Ie	B/d	B/dD	B/a	P/d	P/a	Γ	Mode Description
ν18	132 ± 4		1.9 ± 1.2	**6.7**	**5.7**	**5.1**	**7.3**	**5.6**	a″	τ(CH3)
2ν18	234 ± 4		1.9 ± 1.2						a′	
ν17	312; 315 ^*a*^	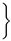	27.3 ± 2.9	**43.7**	**42.1**	**42.5**	**44.6**	**43.7**	a′	δ(C−O−C)
ν16	332 ± 4	a″	τ(C−O−C=O)
2ν16	660		0.2 ± 0.6						a′	
ν15	769		5.1 ± 0.9	**8.6**	6.3	6.3	**8.4**	6.3	a′	δ(O=C−O) −δ(C−O−C)
ν14	928		47.2 ± 2.3	46.4	47.9	47.9	43.2	44.9	a′	ν(C−O) +ν(O−C)
ν13	1024 ± 4		2.9 ± 0.8	3.9	3.8	3.8	4.0	3.9	a″	δop(C−H)
ν12	1167	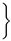	4.3 ± 1	**8.4**	6.4	6.4	**7.0**	4.7	a″	ωop(CH3)
ν11	1167	a′	ν(C–O) −ν(O−C) −ωip(CH3)
ν10	1210		3 ± 0.8	3.9	3.8	3.9	**5.1**	**5.7**	a′	ν(C−O) −ν(O−C) +ωip(CH3)
ν9	1370		10.2 ± 1.1	12.2	9.6	9.6	12.7	10.4	a′	δip(C−H)
ν8	1438	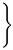	17.1 ± 1.5	23.1	18.5	18.5	**25.2**	20.5	a′	umbrella (CH3)
ν7	1452 ± 4	a″	δas(CH3)
ν6	1465	a′	δs(CH3)
—	1585		1.3 ± 0.8							
$\nu_{5}$	1755		25.5 ± 1.5	**18.4**	24.0	24.0	**18.9**	24.2	$a^{\prime }$	ν(C=O)

**Table 2 molecules-26-04523-t002:** Wavenumbers $\tilde{\nu }$/cm${}^{-1}$ and relative intensities *I* (normalised to 100 in sum) of those fundamentals of methyl methanoate, for which reference values are available [11]. Gas phase reference (r) intensities were corrected for different laser excitation wavelength, their statistical error was estimated from the noise level of the reference data. Gas phase data (gas) from Figure 1. Jet data (jet) from Table 1, integrated jet intensities extrapolated from 20–180 K to 298 K. B3LYP/def2-QZVPP (B/d), intensities adjusted for 298 K. Average values ($\bar{\mathrm{calc}}$) over all computational methods from Table 1, adjusted for 298 K. ^*a*^ Intensity of overtone at 1710 cm${}^{-1}$ was added.

	Gas Reference [11]		This Work							
Mode	${\tilde{\nu }}_{r}$	$I_{r}$	${\tilde{\nu }}_{\mathrm{gas}}$	$I_{\mathrm{gas}}$	${\tilde{\nu }}_{\mathrm{jet}}$	$I_{\mathrm{jet}}$	${\tilde{\nu }}_{B}/d$	$I_{B}/d$	${\tilde{\nu }}_{\bar{\mathrm{calc}}}$	$I_{\bar{\mathrm{calc}}}$
$\nu_{17/16}$	313	13 ± 3	311	32.6 ± 4.6	312	28 ± 4.5	301	40	302 ± 2	38.7 ± 3.2
$\nu_{14}$	919	38 ± 3	926	38.4 ± 4.3	928	39.7 ± 1.8	928	34.6	948 ± 21	33.8 ± 3.1
$\nu_{10}$	1207	6 ± 3	1209	3.1 ± 2.6	1210	2.5 ± 0.6	1229	2.9	1242 ± 14	3.5 ± 0.6
$\nu_{9}$	1368	10 ± 3	1371	8.7 ± 4.5	1370	8.5 ± 0.8	1399	9	1398 ± 1	8.2 ± 1.1
$\nu_{5}$	1751	34 ± 3 ^*a*^	1755	17.1 ± 2.3	1755	21.2 ± 1.2	1790	13.5	1809 ± 21	15.8 ± 2.6

**Table 3 molecules-26-04523-t003:** Individual scaling factors *P* needed to match harmonic theoretical predictions (B3LYP/def2-QZVPP) to the relative experimental Raman intensity (Table 1, Figure 7), normalised to $\nu_{14}$. $P_{I}$ matches integrated intensities, $P_{\sigma }$ matches peak heights for a given Gaussian simulation width σ/cm^−1^. $P_{\sigma }$ that are outside the error margin of $P_{I}$ are marked bold. $P_{h}$ matches calculated Raman cross sections to experimental peak heights, for reference. For $\nu_{17/16}$, the average of $P_{\sigma }\left(\nu_{17}\right)$ and $P_{\sigma }\left(\nu_{16}\right)$ is given.

Mode	$\tilde{\nu }$/cm${}^{-1}$	$P_{I}$	$P_{0.5}$	$P_{1.0}$	$P_{1.5}$	$P_{h}$
$\nu_{18}$	132	0.28 ± 0.17	**0.95**	**0.50**	0.35	0.14
$\nu_{17/16}$	312; 332	0.62 ± 0.07	**0.50**	**0.33**	**0.27**	0.12
$\nu_{15}$	769	0.59 ± 0.10	**1.21**	**0.87**	0.62	0.22
$\nu_{14}$	928	1	1	1	1	1
$\nu_{13}$	1024	0.73 ± 0.20	**1.03**	0.54	**0.37**	0.15
$\nu_{12/11}$	1167	0.51 ± 0.11	**0.77**	0.59	0.47	0.21
$\nu_{10}$	1210	0.75 ± 0.19	0.80	0.81	0.81	1.5
$\nu_{9}$	1370	0.83 ± 0.09	**0.93**	0.91	0.89	1.02
$\nu_{8/7/6}$	1452	0.73 ± 0.07	0.79	**0.41**	**0.29**	0.08
$\nu_{5}$	1755	1.37 ± 0.10	1.40	1.40	1.40	2.22

**Table 4 molecules-26-04523-t004:** Relative energies of minima and transition states (^‡^) of methyl butanoate in kJ mol^−1^ on CCSD(F12*)(T*)//B3LYP level, without ($\Delta E_{\mathrm{el}}$) and with B3LYP ZPE included ($\Delta E_{0}$). In parentheses, the results without CCSD(T) correction are given. Reference values for $\Delta E_{\mathrm{el}}$ on B3LYP/6-31G(d) level [6] and $\Delta E_{0}$ on B2PLYP-D3BJ/aug-cc-pVTZ level [9] added for comparison.

Conformer	$\Delta E_{\mathrm{el}}$		$\Delta E_{0}$		$\Delta E_{\mathrm{el}}$ [6]	$\Delta E_{0}$ [9]
tgt	2.2	(1.8)	3.2	(2.8)	3.1	2.6
tgt–ttt ^‡^	2.3	(1.9)	3.1	(2.7)	–	4.4
ttt	0	(0)	0	(0)	0	0.1
ttt–ttg ^‡^	10.5	(9.9)	10.4	(9.8)	11.3	11.1
ttg	−0.6	(−0.6)	0.0	(0.1)	0.2	0
ttg–tgg ^‡^	5.9	(5.4)	6.8	(6.4)	–	7.2
tgg	1.8	(1.8)	2.9	(2.9)	3.9	2.5
tgg–tgt ^‡^	15.5	(14.7)	16.3	(15.5)	–	–

**Table 5 molecules-26-04523-t005:** Conformational ratio of methyl butanoate in supersonic expansions, determined by different methods. Theoretical populations calculated for relative Gibbs energies ($\Delta G^{Ꝋ}$) or ZPE corrected electronic energies ($E_{0}$) for the temperatures given in parentheses after full relaxation of the second torsional angle.

	Method	ttt	ttg
This	Raman Integrals	43 ± 8	57 ± 8
Work	Raman Peak Heights	46 ± 9	54 ± 9
	CCSD(T)//B3LYP $\Delta G^{Ꝋ}\left(300\;K\right)$	51	49
	CCSD(T)//B3LYP $\Delta G^{Ꝋ}\left(200\;K\right)$	47	53
	CCSD(T)//B3LYP $\Delta G^{Ꝋ}\left(100\;K\right)$	40	60
	CCSD(T)//B3LYP $\Delta E_{0}\left(300\;K\right)$	37	63
Ref. [9]	Microwave	41 ± 4	59 ± 6
	B2PLYP $\Delta G^{Ꝋ}\left(300\;K\right)$	51	49
	B2PLYP $\Delta E_{0}\left(300\;K\right)$	38	62
	MP2 $\Delta G^{Ꝋ}\left(300\;K\right)$	34	66
	MP2 $\Delta E_{0}\left(300\;K\right)$	30	70

## Data Availability

Most of the relevant data are contained within the article or supplementary material. In addition, selected spectra are provided in digital format (.dat) under https://doi.org/10.25625/WITSDA (accessed on 23 July 2021).

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
