# Peer review of "Quantifying Conformational Isomerism in Chain Molecules by Linear Raman Spectroscopy: The Case of Methyl Esters"

_molecules, 2021, doi:10.3390/molecules26154523_

Round 1

Reviewer 1 Report

The manuscript presents a comprehensive analysis of the capabilities of Raman spectroscopy to identify conformers and evaluate their abundances, an important aspect for biosystems.  The strengths and weaknesses of standard gas-phased measurements and supersonic jet expansion are discussed and quantified.  The work is thorough regarding the experimental aspects, with a detailed list of possible limitations, an important reminder for both experimentalists and theoreticians when comparing simulated and recorded data.  The manuscript is quite long but relatively easy to read, and the error estimations well-supported through rational analyses.  The overall work is of high-quality and assuredly recommendable for publications.  I would like the authors to address a few, minor points:

1.  The authors note that diffuse functions in the basis set improves the results, but then choose not to use them because of the cost.  However, the chosen basis set is of quadruple-ζ quality, and the literature has shown that for (simple) hybrid functionals, basis sets of triple-ζ should be sufficient, which should significantly reduce the cost.  It would be useful for the reader to have more explanations on this choice, as the published results seems to show that aug-cc-pVTZ gives very close results.  Moreover, benchmarks in the literature have shown that "lighter" basis sets derived from aug-cc-pVTZ can give very close results, so the inclusion of diffuse functions should be possible without reducing the overall quality, which could be interesting for the larger methyl esters.
2.  The passage "Clearly, the values of this work are more reliable and in particular the jet values may be directly compared to theoretical (by preference anharmonic) predictions. Indeed, in all cases but ν14, the spread of the explored harmonic predictions does not include the experimental value. It remains to be seen whether this gap can be closed by anharmonic correction or whether intrinsic electronic structure errors dominate." on p. 11 is a bit unclear.  The gap seems to refer to theoretical data getting closer to the reference data from Ref. 11, but it is unlikely that the anharmonic correction is so large.  Another possible interpretation would be to get theoretical results closer to the new experimental data, more likely, but not really fitting the reference to "this gap".
3.  The discussion on the anharmonic rovibrational couplings and the reference to Eq. 31 in p. 14 is a bit misleading.  Ref. 31 clearly refers to the rotational couplings in the vibrational levels through the Coriolis couplings, but not the rotational levels. The reference to the low-energy, large amplitude motions (Ref. 13) is also unexpected, since it refers to issues in the description of the potential energies related to strongly anharmonic modes as fourth-order polynomials, which is further aggravated in the VPT2 formulation, again on vibrational levels.  In the context of rotational transitions, better references should be the treatment of mixed rotation-vibration couplings, for instance as done for the prediction of far-IR spectra.
4.  MN15 is a semi-empirical exchange-correlation functional, like B3LYP.  Semi-empirical level can refer to the actual methods, which have nothing to do with DFT.  I would recommend the authors to change the text in line 597-598 to avoid any confusion and make clear that they are referring to the exchange-correlation functional.
5.  The keyword "anharmonicity" is ill-suited.  This keyword commonly refers to the theoretical model (within the Born-Oppenheimer and Eckart conditions), and is not considered in this manuscript.
6.  The order in the ESI does not follow the text. Since the text simply refers to the ESI at times with no specific section, it would be easier to follow the same order as the manuscript, or give more precise indications to the reader in the body of the manuscript.

a.  l.581-582: "the harmonically ZPE (zero point vibrational energy) corrected minimum is higher than the ZPE-corrected barrier (without the imaginary contribution) to ttt," should be reworded.
b.  l. 630: "larger distance to the nozzle" -> "longer distance from the nozzle"
c.  In the code in ESI 5.1 there is a small typo "abbreviaiton"

Reviewer 2 Report

see attached file

Reviewer 3 Report

In Figure 7 could an inset be provided or something similar to Figure 6 such that it is easier to examine weaker peaks and how well the simulations match the experimental data?

What is the significance of the question mark in the middle panel of figure 12, is that because the harmonic calculation predicts 2 signals? Further the color scheme if the middle panel is hard for me to interpret, what is the pink color of the peak at 875, is that due to the red and blue curves overlapping each other?

Reviewer 4 Report

The manuscript is clearly developed and provides an important set of data in the study of the quantification of conformational isomerism in methyl esters. I would suggest its acceptance in its current form. However, I respectfully suggest to the authors emphasize the objective of the article in the introduction and its impact already diluted in the generous amount of data from the manuscript.
